# Identifying community values related to heat: recommendations for forecast and health risk communication

Kathryn Lambrecht[1], Benjamin J. Hatchett[2], Kristin VanderMolen[2], Bianca Feldkircher[3]

[1]Technical Communication, Arizona State University Polytechnic, 7001 E Williams Field Rd., Mesa, AZ, 85212, USA

[2]Western Regional Climate Center, Desert Research Institute, 2215 Raggio Parkway, Reno, NV, 89512, USA

[3]National Weather Service, NOAA, 2727 E Washington St., Phoenix, AZ, 85251, USA

*Correspondence to*: Kathryn Lambrecht (Kathryn.Lambrecht@asu.edu)

**Abstract.** Effective communication of heat risk to public audiences is critical to promoting behavioral changes that reduce susceptibility to heat-related illness. The U.S. National Oceanic and Atmospheric Administration (NOAA) National Weather Service (NWS) provides heat-related information to the public using social media platforms such as Facebook. We applied a novel rhetorical framework to evaluate five years (2015–2019) of public responses to heat-related Facebook posts from the NWS office in Phoenix (Arizona) to identify "commonplaces" or community norms, beliefs, and values that may present challenges to the effectiveness of heat risk communication. Phoenix is in one of the hottest regions in North America and is the tenth largest metropolitan area in the U.S. We found two key commonplaces: 1) the normalization of heat, and 2) heat as a marker of community identity. These commonplaces imply that local audiences may be resistant to behavioral change, but they can also be harnessed in an effort to promote protective action. We also found that public responses to NWS posts declined over the heat season, further suggesting the normalization of heat and highlighting the need to maintain engagement. This work provides a readily generalizable framework for other messengers of high-impact weather events to improve the effectiveness of their communication with receiver audiences.

**1 Introduction**

Extreme heat is the deadliest weather or climate-related hazard in the U.S. (National Weather Service 2019). Heat-related mortality is exacerbated by the persistence of knowledge gaps between messengers (e.g., weather forecasters and public health officials) and receiver audiences that limit the effectiveness of heat risk communication and/or delivery (Abrahamson et al. 2007; Chowdhury et al. 2008, 2012; Semenza et al. 2008). Yet, where heat risk communication has been effective (and barriers to self-protection have been limited, see Toloo et al. 2013), it has been shown to lower mortality (Ebi et al. 2004; Schifano et al. 2012). Often research aimed at improving the effectiveness of heat risk communication therefore has focused on closing messenger-receiver knowledge gaps. Recommended approaches for doing so include creating virtual public education opportunities (Neumann et al. 2018; Stephens et al. 2019), increasing the consistency and repetition of messaging (Hawkins 2017; Keul 2018), and integrating nuanced understanding of public knowledge and beliefs into risk communication (Chowdhury et al. 2012). However, to accomplish the latter, frameworks and tools are needed to help communicators of risk identify that nuance.

The identification of "commonplaces" (referred to as *topoi* in the field of rhetoric, originating in the work of Aristotle) offers one approach for building place-based nuance into risk communication. Commonplaces are the underlying norms, beliefs, and values that influence the way audiences respond to information (Walsh and Boyle 2017), including about weather and safety (Lambrecht et al. 2019). The identification of commonplaces therefore enables communicators of risk to acquire important rhetorical awareness of their unique target audiences (Walsh and Boyle 2017) and better tailor their communication for increased effectiveness, for example by presenting content framed specifically within local norms and values (Lambrecht et al. 2019; see Chowdhury et al. 2012). As the identification of commonplaces is a practice grounded in analyzing language, the method can be applied to any technical or risk communication (Harlow 2015).

Here we identified commonplaces related to the communication of heat risk in Phoenix, Arizona, through analysis of public comments posted in response to National Oceanic and Atmospheric Administration (NOAA) National Weather Service (NWS) heat-related forecasts on Facebook between 2015–2019. We additionally evaluated the frequency and type of public comment in relation to temperature anomalies using a percentile-based approach. Our findings confirm perceptions held within the NWS Phoenix office and offer insights into how communicators of heat risk might leverage commonplaces to tailor the content and timing of messages for increased effectiveness. The methods utilized offer a replicable approach for communicators of heat risk in any location, or of different weather and climate-related hazards (e.g., winter weather; Lambrecht et al. 2019), to acquire the nuanced understanding of public knowledge and beliefs encouraged for closing messenger-receiver knowledge gaps (see Chowdhury et al. 2011).

**2 Methods**

To identify commonplaces related to the communication of heat risk in Phoenix, we collected and analyzed 4,304 public comments from all NWS Phoenix Facebook posts that discussed heat-related forecasts or above-average temperatures during the warm season (June-September) between 2015-2019. The comments were expressed either in written text or as visual

objects (e.g., GIFs, photographs, or memes). We chose to focus our analysis on NWS Phoenix because its County Warning Area (CWA) is subjected to extreme heat. The warm season months of June-September were selected for analysis because the vast majority (94%) of heat-related deaths in the U.S. occur during that time of year (CDC 2017). The period of focus, 2015-2019, was selected to capture enough data to allow for the identification of patterns in commonplaces over time.

Following Lambrecht et al. (2019), the lead author inductively coded a subsample (~728 or ~17%) of the total 4,304 public comments according to how they functioned as a response to heat-related NWS Phoenix Facebook posts (see also Walsh and Ross 2015). The resulting codebook was shared and discussed with co-authors, who agreed that it thoroughly and accurately reflected the different types of responses (see Saldaña 2015). In total, the codebook included thirteen codes (i.e., different types of comments): (1) *feelings or reactions* shared in response to an NWS Facebook post, (2) *tags* linking the name of a person to a post to draw their attention to it, (3) *verifications*, or sharing information that confirms or refutes a forecast, (4) *comparisons* differentiating weather in Phoenix from another location, (5) *questions* asking about heat impacts, (6) *commentaries* exploring the political context of heat, (7) *past experiences* or sharing stories about heat, (8) *appeals to safety* or warning other members of the public about heat, (9) *information* sharing (e.g., about resources), (10) *changes in plans* indicating that weather played a role in modifying activity, (11) *thank yous* expressing appreciation for NWS, (12) *advice* sharing between members of the public, and (13) *requests* asking for additional weather information or changes in weather (Table 1). The lead author then analyzed the comments within and across each of those categories to identify commonplaces related to heat. For a particular norm, belief, or value to have been considered a commonplace, it had to be: (1) expressed across multiple years and by multiple people, 2) related directly to Phoenix (as opposed to other locations), and 3) made in response specifically to an NWS Facebook post on heat (as opposed to weather more generally) (see also Walsh and Boyle 2017). After coding ~3,637 (or ~85%) of the total 4,304 public comments, the lead author met with co-authors to share and discuss the potential commonplaces that had emerged and to evaluate whether they met the above three criteria (see Saldaña 2015). There was consensus among all authors about which potential commonplaces did and did not meet the criteria. Those that did not were excluded from the analysis.

To evaluate the frequency and type of public comment in relation to temperature anomalies, we utilized daily minimum and maximum temperature data as well as archived excessive heat warnings issued by NWS (https://www.weather.gov/psr/heat). Quality controlled data for the first order station at the Phoenix Airport (period of record: 1933–2019) were acquired from the Applied Climate Information Services website (http://xmacis.rcc-acis.org/). Percentile-based approaches are commonly used to study extreme heat events (Meehl and Tebaldi 2004; Perkins and Alexander 2013). Following Shortridge et al. (2019), percentiles for minimum and maximum temperature were estimated using an eleven-day moving window centered on the day of interest for the period of record between May 15–October 15 to remove seasonality effects (Montecinos et al. 2017). We defined heat events when maximum and/or minimum temperatures exceeded the 95th percentile.

**3 Results**

Of the thirteen categories (or types) of comments identified (Table 1), the most common was the sharing of feelings or reactions (48%). Tags was the second most common (14%) followed by comparisons (8%). Our analysis of the comments within and across each of the total thirteen categories revealed two salient commonplaces: (1) the normalization of heat (i.e., that heat is "normal" or to be expected), and (2) heat as a marker of community identity (i.e., that the ability to withstand heat is part of being Phoenician). Both commonplaces validate perceptions held within the NWS Phoenix about the community norms, beliefs, and values shared among members of the local public.

Table 1. Summary of public comment categories (or types)

| Category (or type) of public comment | Example | Phoenix, AZ (n=4304 comments on 345 posts) | |
|---|---|---|---|
| | | Number of posts | Percent of total |
| **Feelings/reactions** | *"I'm dying of heat."* | 2059 | 48% |
| **Tags** | *"Grandma, check out this forecast!"* | 623 | 14% |
| **Comparisons** | *"This heat is why I moved to Monterey."* | 350 | 8% |
| **Verifications** | *"116° in my car today, and it's only 10 am."* | 247 | 6% |
| **Questions** | *"When will the monsoon get here?"* | 195 | 5% |
| **Commentary** | *"Will the city turn off sprinklers and stop issuing building permits? No, while the mayor symbolically jumps on the climate change bandwagon"* | 178 | 4% |
| **Past Experiences** | *"I'm old enough to remember when Monsoon season started when we had 3 consecutive days with dew points at or above 55"* | 174 | 4% |
| **Appeals to safety** | *"Keep your dogs inside!"* | 152 | 4% |
| **Information** | *"Check the website for the Flood Control District of Maricopa County. They have maps with rain gauges all over the county."* | 109 | 3% |
| **Changes in plans** | *"Guess I'll be rethinking the zoo today."* | 88 | 2% |
| **Thank yous** | *"US National Weather Service Phoenix Arizona you're doing a great job with us cranky and dried out Phoenicians. Thank you!"* | 58 | 1% |
| **Advice** | *"Enjoy this. The heat is coming!"* | 36 | <1% |
| **Requests** | *"We are tired of the heat and WOULD LIKE SOME RAIN"* | 35 | <1% |

Evidence for the normalization of heat was most prevalent in the sharing of feelings or reactions. Often, the feelings expressed were of frustration about heat but with resignation toward or acceptance of its perceived normalcy. For example, as

expressed in the following public comments: "Arizona is that feeling when you open an oven to check on your cookies and it burns your face except there's no cookies and you can't escape" and "I get it, this is the life I made for myself" (see also Fig. 1). Other evidence for the normalization of heat was found in feelings or reactions that conveyed a lack of distinction, or at times possibly a sense of confusion, about maximum temperatures. This lack of distinction and/or sense of confusion is illustrated, for example, in the following public posts: "After it reaches 100[°F], what really is the difference? All we can say here is, 'but it's a dry heat' ha!" and "What makes 113[°F] excessive heat but 109[°F] not?" Finally, the few questions asked about heat and its potential health risks, appeals to safety, and changes in plans relative to the prevalence of feelings or reactions that expressed resignation may further suggest that having been normalized, warnings about heat are not received with concern.

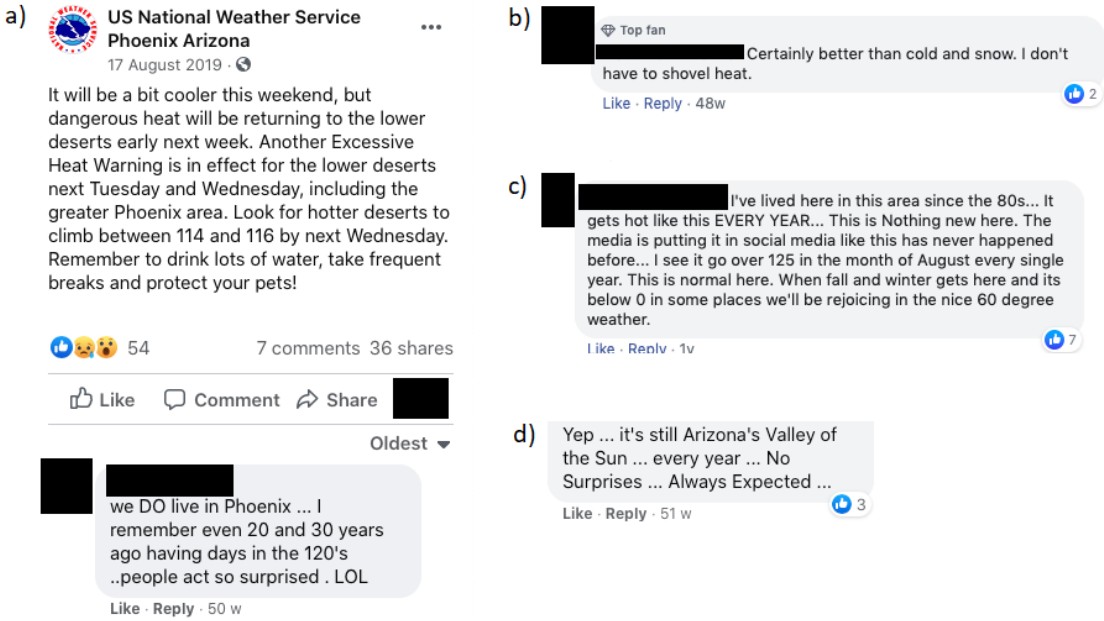

Figure 1. (a-d) Comments posted by the public in Phoenix reflecting the normalization of (and resignation toward) heat.

The second commonplace, also most prevalent in feelings or reactions, is the role of heat as a marker of community identity, wherein the ability to withstand heat is communicated with pride and used as a means of inclusion (e.g., "Records are meant to be broken!"). In contrast, the inability to withstand heat is sometimes met with rebuke and used as a means of exclusion (e.g., "If you can't stand the heat, get out of Arizona!" or "Even my dog is used to it. Hot schmot. Get over it."). Such sentiments are especially common in memes (not shown here due to copyright protection) depicting movie characters in battle with accompanying text like, "Stop whining about the heat, we are Arizonians!" and "You merely adopted the Arizona heat. I was born into it. Molded by it." Though the normalization of heat and heat as a marker of community identity are two distinct commonplaces, the latter does also offer evidence for the former as notions of "community" are built in part around norms (i.e., in this case, that heat is "normal") (see Lidskog 2018).

NWS Phoenix posted at least once on a total of 231 days between 2015–2019 (Fig. 2), with all excessive heat warnings corresponding to an NWS post on the same day or in the day prior. Posting frequency increased throughout the study period by 10 posts per year, from 26 in 2015 to 66 in 2019. A single post occurred on 148 days, and 83 days involved multiple posts.

Multiple post days increased from 3 in 2015 to 35 in 2019, while single posts grew from 23 in 2015 to a maximum of 39 in 2018. Comparing days with multiple and single posts with a two-sample Student's t-test, multiple post days generated significantly more engagement in terms of public reactions and comments (p=4.39e-15 and p=1.52e-8, respectively), but not shares (p=0.102). Public engagement was inversely correlated with day of year (Spearman's correlation = -0.46, p = 1.38e-07), implying that responses were more frequent earlier in the warm season (e.g., June-July) than later (August-September).

Nonetheless, extreme heat events generated responses even later in the year (e.g., September 2017 and August 2019). Any time extreme heat occurred, NWS posted at least once on Facebook. In several cases, NWS began posting several days before extreme temperatures (Fig. 2), eliciting the largest number of responses between one and three days before the extreme temperatures were observed. The June 16, 2017 posts generated the most single-day responses (8,455) with the July 22, 2018 event including 3,056 responses. Both heat events involved three days where daily maximum temperature records were set.

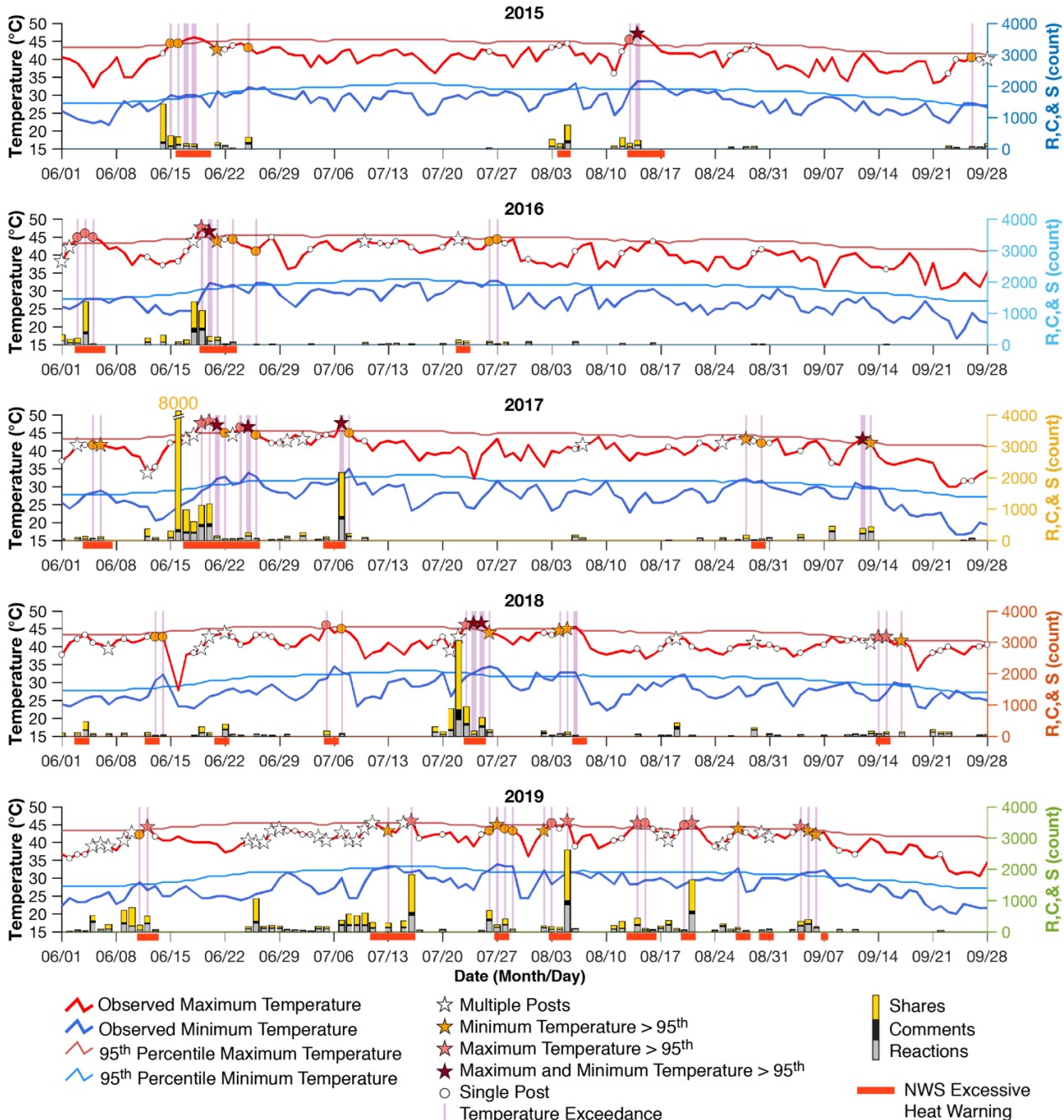

Figure 2. Phoenix Airport temperatures (left y-axis) and Facebook engagement (right y-axis; reactions (R), comments (C), and shares (S) for the NWS Phoenix spanning the June–September periods for (a) 2015, (b) 2016, (c) 2017, (d) 2018, and (e) 2019. Excessive heat warnings are shown by red bars along the x-axes.

## 4 Discussion

### 4.1 The normalization of heat

Rather than asking questions about the heat and its potential health risks, or alerting others to those risks, public comments in response to NWS Phoenix heat-related forecast posts were more likely to communicate a feeling or reaction, often of frustration and/or resignation toward the perceived normalcy of heat. However, the normalization of heat is not unique among this audience. For example, Abrahamson et al. (2008) found that some individuals in the UK did not perceive heat to be a risk to their health specifically because of their prolonged exposure to it over time. This raises important questions about how to encourage those "accustomed" to heat to protect themselves from its potential health risks. One strategy might be to add more texture to the communication of heat risk, for example by using more clearly differentiated terms than the commonly used descriptors of "excessive heat," "dangerous heat," and "record heat." Another strategy might be to offer more detail, for example about what risks could be present (or otherwise increase in likelihood of occurrence) at what temperature ranges and for which groups. Lastly, an additional strategy might be to emphasize the real potential specifically of *local* heat risks, for example by way of links to local news stories documenting incidents of heat-related illness. Appendix A contains a recent example of how NWS Phoenix has begun to apply these strategies (Fig. A1). It also includes examples suggested by the authors (Table A1). It is important to note that any new messaging should additionally strive to be consistent with risk communication best practices (e.g., Li et al. 2021; Lindell and Perry 2012; Mileti 2000; NOAA 2016).

### 4.2 Heat as a marker of community identity

Heat as a marker of community identity is a strong commonplace. Although in Phoenix people were more likely to react with or express feelings of frustration about the heat, they often conveyed a related sense of pride, solidarity, and community membership in their ability to withstand it. Such sentiments were particularly pronounced in the memes posted by the public which, research has shown, can contain powerful messages of community norms and expectations (Dancygier 2017; Kahan 2017; Ross and Rivers 2019). Simply cautioning a public that prides itself on withstanding the dangers of heat therefore may not work to convince people that they are truly at risk and should adopt behavioral changes to keep themselves safe. Rather, carefully drawing on and possibly reframing signals of community membership as conveyed in public posts may offer benefits for encouraging acceptance of risk and enactment of protective behavior (see also Lambrecht et al. 2019). For example, reframing heat safety as a community norm, or communicating that protection against heat builds a stronger community, could be potentially effective ways to utilize sense of belonging to encourage safety (Table A1).

### 4.3 The timing of messaging

The decline in public responses with time during warm seasons (Fig. 2) supports the normalization of heat also as the year progresses. However, several occasions where responses increased late in the year provide guidance to increasing community engagement. The greater frequency of NWS posts and responses during 2019 highlights additional strategies to maintain community engagement. For example, using creative ways – following best practices – to communicate the same

information (e.g., through infographics and videos) or highlighting records may serve as mechanisms to attract public attention

(Dunlap and Lowenthal 2016; Lazard and Atkinson 2014). Lastly, although NWS Facebook posts typically were coincident with extreme heat and overlapped with or preceded excessive heat warnings, there is evidence to suggest posting several days in advance and posting multiple times per day ("early and often") offers actionable information for the NWS to provide the public with an early heat warning system.

**5 Conclusion**

The identification of commonplaces can serve as one approach to closing knowledge gaps between communicators of risk and receiver audiences (the public). Our analysis of public comments in response to heat-related forecasts on the NWS Phoenix Facebook site validated NWS forecaster beliefs and revealed two predominant commonplaces: 1) the normalization of heat, and 2) heat as a marker of community identify. These commonplaces reflect norms, beliefs, and values that may present challenges to the effectiveness of heat risk communication. They imply discouragement of protective action, but they

can also be harnessed in an effort to promote positive behavioral change. A promising direction of future research is to explore whether the above described or other recommendations can help to reorient norms, beliefs, and values *toward* encouragement of protective action. Additional research into the prevalence of these commonplaces across other regions characterized by extreme heat will provide important testbeds for the potential transferability of newly crafted heat risk communication to other regions. Delivery strategies that may also increase the effectiveness of heat risk communication include messaging campaigns

beginning several days in advance of a forecast of extreme heat which, especially if they leverage commonplaces, might garner more attention. Later in the season, focused posts that attract attention (e.g., records, comparisons with other locales) may also serve to communicate heat safety strategies while providing interesting information to the community. Such endeavors related to the content and timing of messaging will only gain in importance and value as heat risk and population exposure to extreme heat continue to increase.

**Appendix A**

The below NWS Phoenix Facebook post from September 9, 2021, includes examples of messaging informed by the commonplaces identified. Specifically, the normalization of heat is countered in the first paragraph, and heat as a marker of community identity is reframed in the second.

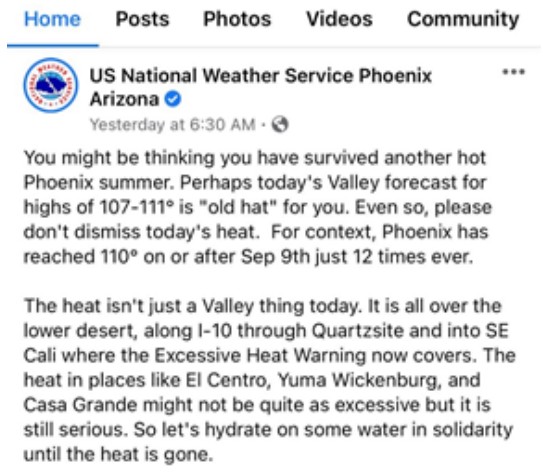

Figure A1. Example of NWS Phoenix heat risk messaging informed by the commonplaces identified (September 9, 2021)

The below table contains examples suggested by the authors for informing messaging using the commonplaces identified. The table is organized by each commonplace and the set of corresponding recommendations provided in sections 4.1 and 4.2.

Table A1. Suggested examples of heat risk messaging informed by the commonplaces identified

| Commonplace | Recommendation | Example |
|---|---|---|
| **The normalization of heat** | Offer more detail around commonly used terms, like "excessive heat" and "record heat" to enhance clarity. | Another day of heat brings another day of caution. Today's excessive heat warning means potential for extremely dangerous conditions within the next ~12 hours. So please prepare now to protect yourself, your family, your neighbors, and your pets. Here's how: Surviving Arizona Heat.

Today's record heat is going to be hot enough to pose potential health risks. So remember to stay in air-conditioned areas if possible (and/or rest often in shade), drink water even if you don't feel thirsty, and avoid strenuous activity if you can. For more advice on how to stay cool: Surviving Arizona Heat. |
| | Offer more detail about what risks could be present and/or which groups might be more susceptible. | Even after a long summer, heat STILL poses potential health risks. Children, pregnant women, athletes, outdoor workers, older adults (65+), and people with medical conditions (like heart disease or high blood pressure) are especially susceptible. |

| | | Even if you've "survived" the heat before, you may not have been unaffected by it. It's important to know how to recognize the symptoms of heat cramps, heat exhaustion, and heat stroke. Here's how: Surviving Arizona Heat. |
| --- | --- | --- |
| | Emphasize the real potential specifically of local heat risks. | Phoenix experiences a LOT of heat, but it isn't unaffected by it. There were 323 heat-related deaths in Maricopa County alone last year (2020). So please take care of yourself this summer - it's likely to be another hot one. Here are some tips you can follow to stay cool: Surviving Arizona Heat. |
| | | Suspect you might be "accustomed" to the heat by now? In 2020, 63% of heat-related deaths in Arizona occurred among people who had lived here for 20+ years. So please take today's high temperatures seriously, regardless of how long you've lived here. |
| **Heat as a marker of community identity** | Reframe heat safety as a community norm. | Pets are a part of our community, too! If you're planning on walking yours today, consider doing so during the coolest part of the day (4:00 am - 7:00 am). And please never leave them in the backyard or a hot vehicle. |
| | | As we hit a new high today, let's all do our part to keep our communities safe! Please help spread the word AND this map of Maricopa County cooling stations and water donation sites. |
| | Communicate that protection against heat builds a stronger community. | Contrary to popular belief, even Arizonians are susceptible to heat! Prolonged exposure can result in heat cramps, heat exhaustion, and heat stroke. Help build a safer and stronger community by sharing these tips for staying cool: Surviving Arizona Heat. |
| | | Interested in helping to make your community stronger? Check in with your family, friends, and neighbors to make sure they're aware of this week's heat wave and how to be prepared for its potential health risks, like heat cramps, heat exhaustion, and heat stroke: Surviving Arizona Heat. |

**Code availability:** Analysis code for the Phoenix data in MATLAB is available upon request.

**Data availability:** All data is publicly available.

**Author contribution:** KL, BJH, and KV conceptualized and designed the research. KL and BJH performed the analyses. KL, BJH, and KV prepared the manuscript with critical input from BF.

**Competing interests:** The authors declare that they have no conflicts of interest.

**Acknowledgements:** Funding for this work was provided by the National Oceanic and Atmospheric Administration International Research Applications Project under agreement A18OAR4310341.

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
