# Peer review of "Identifying community values related to heat: recommendations for forecast and health risk communication"

_Geoscience Communication, 2021_

## Author Response (AR1)

Dear Dr. Bohan (Reviewer 1)
We thank you again for your kind comments and constructive suggestions for improvement. We have updated our initial responses to your comments in-line below.

In the manuscript "Identifying community values related to heat: recommendations for forecast and health risk communication" the authors outline how they analyse and evaluate public comments on the NWS Facebook page related to heat risk in Phoenix, AZ. They determine that there are 2 key "commonplaces", or community norms or values, that describe and challenge heat related threat assessment in Phoenix - "the normalization of heat" and "heat as a marker of community identity." They then describe ways that these commonplaces make the communication of heat related messaging difficult, as well as ways to utilize these commonplaces in future messaging. The nuances in understanding an at-risk populations perception of a hazard is critical to effectively messaging about that hazard in such a way that people are motivated to take protective action. By identifying some of these critical nuances the authors are providing an important roadmap to communicators about how to message more appropriately and effectively.

The approach that they take to evaluate the Facebook messages is robust, and their description of the issues is clear and sound. However, the manuscript would be of greater value to communicators looking to improve messaging using these findings if the authors include additional information and suggestions around how to utilize the identified commonplaces when creating heat warning messaging campaigns. For instance, they rightly suggest that additional language should be used besides "excessive heat" etc. What might be more appropriate or descriptive terms? Can they offer a more extensive list of potential suggestions as to how to reframe the commonplace of community membership to inspire people towards action?

We appreciate the recommendation to provide examples of how communicators can use the paper's findings to improve messaging. We have added an appendix to the paper that contains examples developed and used by NWS Phoenix (Figure A1) as well as additional examples suggested by the authors (Table A1).

Additionally, were there any examples of messaging that was found to inspire action through changing the commonplace narratives that could be used as a template?

Our analysis concluded with the identification of commonplaces and did not extend into their potential use (and public response). Since receiving the above reviewer comment, NWS Phoenix has begun to inform messaging with the commonplaces identified (for example, in Figure A1). However, our review of public responses to those messages suggests that it is likely too early to know whether/how they might inspire action. That said, this would be an interesting direction of future research.

Regardless, a list or table of suggested messaging could be very helpful for science communicators tasks with creating copy around dangerous heat events.

We agree and have included examples from NWS Phoenix (Figure A1) and from the authors (Table A1) in an appendix.

Although it may be beyond the scope of this work it would be interesting to hear the authors thoughts on the efficacy of different types of hazard messaging on social media - are videos, infographics, photos or articles more effective for communicating risk to different populations?

We revisited the data and have determined that, unfortunately, it doesn't allow us to comment confidently on which types of messaging might be most effective. The lack of identifiable demographic information also prohibits us from making any links between types of messaging and receiver characteristics. However, we have added references (pg. 9, line 229) to direct readers to resources for developing effective messaging types. We have also added references (pg. 8, line 202) to direct readers to resources on risk communication best practices more generally.

Dear Reviewer 2,
Our thanks again for your positive and helpful feedback. We have updated our initial responses to your comments in-line below.

This is a well-written paper on a topic of growing importance. As extreme heat events become more frequent across the world, we should be taking lessons from places like Phoenix where extreme heat is already commonplace and applying them more generally for risk communication. I have only a few suggestions for revision, as follows:

- Introduction: I'd like to call the authors' attention to a recently published paper in Weather, Climate, and Society that analyzed Twitter posts on extreme heat from NWS accounts, which would be a relevant citation in the literature review: https://doi.org/10.1175/WCAS-D-20-0039.1

We appreciate the reviewer alerting us to this publication and it has been added as a reference to our revised discussion (pg. 8, line 202). The findings have also informed the examples we have added in the appendix (Table A1).

- Methods: I'd suggest providing more information on the coding process: how many coders were involved, and was intercoder reliability measured?

We have added additional information on the coding process. Specifically, we have explained that the lead author was responsible for the development of the codebook, which was shared and discussed with co-authors prior to being finalized. We have also explained that the lead author analyzed the public comments within and across the code categories, and that the commonplaces identified were shared and discussed with co-authors to ensure that they met the criteria for being considered a commonplace. If there is additional (or specific) information on the coding process that would be of benefit to include, we are happy to provide it.

Minor suggestions:

- Page 2, line 32: I'd suggest specifying that extreme heat is the deadliest hazard in the US, which is what the cited reference supports.

Thank you for this suggestion. We have modified the text to include "in the US."

- Abstract, line 15: possible typo, should be "community identity"

We have corrected this typo. Thank you for catching it.

Note: In addition to addressing the above comments and after our own review of the paper, we have added a few very minor edits to the text to improve clarity. We have also updated Figure 2 with a few minor edits to the legend. These include correcting the star color order, moving the "th" in "95th" to superscript, and adding "NWS" to "Excessive Heat Warning."